# Dental Pulp Stem Cell-Derived Secretome and Its Regenerative Potential

**DOI:** 10.3390/ijms222112018

**Published:** 2021-11-06

**Authors:** Julia K. Bar, Anna Lis-Nawara, Piotr Grzegorz Grelewski

**Affiliations:** Department of Immunopathology and Molecular Biology, Wroclaw Medical University, Bujwida 44, 50-345 Wrocław, Poland; anna.lis-nawara@umw.edu.pl (A.L.-N.); piotr.grelewski@umw.edu.pl (P.G.G.)

**Keywords:** dental stem cells, secretome, paracrine effect, regenerative medicine

## Abstract

The therapeutic potential of the dental pulp stem (DSC) cell-derived secretome, consisting of various biomolecules, is undergoing intense research. Despite promising in vitro and in vivo studies, most DSC secretome-based therapies have not been implemented in human medicine because the paracrine effect of the bioactive factors secreted by human dental pulp stem cells (hDPSCs) and human exfoliated deciduous teeth (SHEDs) is not completely understood. In this review, we outline the current data on the hDPSC- and SHED-derived secretome as a potential candidate in the regeneration of bone, cartilage, and nerve tissue. Published reports demonstrate that the dental MSC-derived secretome/conditional medium may be effective in treating neurodegenerative diseases, neural injuries, cartilage defects, and repairing bone by regulating neuroprotective, anti-inflammatory, antiapoptotic, and angiogenic processes through secretome paracrine mechanisms. Dental MSC-secretomes, similarly to the bone marrow MSC-secretome activate molecular and cellular mechanisms, which determine the effectiveness of cell-free therapy. Many reports emphasize that dental MSC-derived secretomes have potential application in tissue-regenerating therapy due to their multidirectional paracrine effect observed in the therapy of many different injured tissues.

## 1. Introduction

In the last decade, stem cells have been proposed for use in regenerative medicine, with attention focused on mesenchymal stem cells (MSCs) as a promising therapeutic option in regenerative medicine and tissue engineering due to their regenerative and protective abilities [1]. The initial therapeutic use of MSCs in regenerative medicine was a cell-based approach, in which stem cells were administered directly to the injured tissue and differentiated into different functional cell types, leading to tissue repair and regeneration [2,3,4,5]. The second approach was tissue engineering based on combining stem cells or differentiated cells with a biodegradable scaffold to form a tissue structure [6,7]. A majority of clinical trials have been performed using MSCs aspirated from bone marrow [8]. It is worth underlining that MSCs isolated from human dental pulp tissue have also been used in clinical trials as a tool in regenerative medicine [9]. Clinical data continue to show successful applications of MSCs, and the completed trials have been listed at www.clinicaltrials.gov (accessed on 1 November 2021), the database of The National Institutes of Health [8,10]. Reports also show that MSCs obtained from different sources are effective in tissue regeneration, but there are still certain important obstacles that limit their application in regenerative medicine [8,11,12]. Most clinical trials use a similar procedure to prepare MSCs for therapy. Depending on the source, stem cells are aspirated from bone marrow or isolated from adipose or pulp tissue and cultured using an optimal medium inducing MSC proliferation [8,9]. Manufacturing should take place in a closed and aseptic environment. The MSCs should be obtained and manipulated following good manufacturing practice (GMP) [13,14,15]. The guidelines of the regulatory agencies, established procedures, and strict safety measures should be followed, and appropriate equipment, reagents, and supplies should be used [8,10]. In the EU, GMP production is regulated under European Regulation No. 1394/2007. In the US, GMP human MSC production is regulated by FDA CFR Title 21, part 1271, subpart D, sections 145e320, focusing on current good tissue practice requirements, which provide exemptions, maintenance of quality procedure, process control, and validation [16]. All current clinical trials have recognized the importance of developing standardized protocols that comply with international regulations to ensure the effectiveness and safety of the treatment [8,9]. In light of these stipulations, potential novel therapies have emerged in regenerative medicine and tissue engineering [4,8,14,17].

Recently, growing evidence has indicated the importance of paracrine signaling induced by MSCs as a supportive mechanism for the regeneration of damaged tissue [8,10,14,17]. Experimental studies showed that the repair effect observed in MSC grafts on the injured tissue was mainly caused by the secreted factors released by the stem cells, rather than cellular engraftment, in an animal model [15,18,19]. Some studies also showed that the differentiation capability of MSCs was not the predominant mechanism in repairing tissue damage in most diseases, but tissue reparative properties have also been attributed to the bioactive factors secreted by the MSCs contributing to the paracrine activity [8,11,14,17]. Due to issues associated with MSC-based therapy, cell-free therapy using the MSC-secretome and their paracrine action instead of cellular replacement or cell–cell interrelation may be a better option as a new therapeutic opportunity in regenerative medicine [1,8,10,11,17,20]. Although the mechanism of the MSC-derived secretome in regeneration is still not fully understood, there are many publications on the MSC-secretome therapeutic potential; however, there are few studies on the dental MSC-derived secretome used in regenerative medicine and tissue engineering. In this review article, we discuss the usefulness of the human dental pulp stem cell secretome in cartilage, bone, and neural tissue regeneration.

## 2. Mesenchymal Stem Cells

MSCs are undifferentiated cells known for their self-renewal and differentiation properties. Bone marrow-derived MSCs (BMMSCs) were the first to be discovered and well documented. Human BMMSCs are a fibroblast-like population, adhere to plastic and express cluster of differentiation (CD) markers (CD10, CD13, CD44, CD73, CD90, CD105, CD106, CD146, CD166, STRO-1) but the stem cells lack the expression of CD11b, CD14, CD19, CD34, CD45, CD79a, and Human Leukocyte Antigen HLA-DR [1,3,16,21,22]. These stem cells express additional surface markers such as platelet-derived growth factor receptor (PDGF-R), epidermal growth factor receptor (EGF-R), insulin-like growth factor receptor (IGF-R), CD49a/CD29, nerve growth factor receptor (NGF-R) [21,22]. Human BMMSCs typically express major histocompatibility complex MHC-I but lack expression of MHC-II, CD40, CD80, and CD86 on the cell surface, and thus they escape T cell recognition [1,5]. BMMSCs have a high proliferative capacity, life span, differentiation potential and immunomodulatory ability [16]. BMMSC are able to self-renew, support hematopoiesis and possess multi-lineage differentiation potential and differentiate into osteoblasts, adipocytes and chondroblasts but also can differentiate into tendons and myogenic and neural cells [3,5,16,22]. Due to the pain and morbidity accompanying MSCs obtained from bone marrow, alternative donating organs have been sought as their source [8,21]. Stem cells with the biological MSC characteristics of other adult tissues were collected from various tissues, such as bone marrow [1,21], placental tissue [23,24], cord blood [23], umbilical cord tissue [24], adipose tissue [23], or the dental pulp of permanent and deciduous teeth [25,26]. Many studies focus on the isolation of MSCs from more accessible sources, aiming for a safe clinical application of these cells in regenerative medicine and tissue engineering [21,27,28]. Consequently, dental-related tissues are proposed as one of the most promising non-invasive sources of human stem cells [6,12,21,27]. Stem cells isolated from different tissues must display the features of MSCs [6]. The International Society for Cellular Therapy (ISCT) established three minimal criteria for MSCs, i.e., (a) they must be adherent to plastic under standard culture conditions; (b) they must express specific markers, such as CD105, CD73, and CD90, and lack the expression of CD45, CD34, CD14, CD11b, CD79alpha, and CD19 and HLA-DR surface molecules; and (c) they must have the ability to differentiate into osteocytes, chondroblasts, and adipocytes [29]. In 2016, ISCT modified the criteria for MSCs and added three new ones, specifically, quantitative RNA analysis of selected gene products, flow cytometry analysis of functionally relevant surface markers, and a protein-based assay of the secretome [30]. Several studies have demonstrated that the MSCs derived from different tissues exhibit heterogenous biological characteristics and functional features due to differences in their differentiation potential, proliferative activity, and immunomodulatory and protective potential, even though they display a similar immunophenotypic profile [2,22,31]. Currently, there is growing evidence that dental stem cells (DSCs) have many similarities to BMMSCs and can be easily isolated, thus having a clear advantage over the costly and invasive techniques required with MSCs collected from bone marrow [6,13,21,32,33]. Dental stem cells can be isolated from various dental soft tissues, human dental pulp tissue of permanent teeth (hDPSCs), human exfoliated deciduous teeth (SHEDs), apical papilla (SCAPs), dental follicle tissue (DFPCs), periodontal ligament (PDLSCs), and gingiva tissue (GMSCs); hDPSCs, SHEDs, and SCAPs are referred to as dental pulp-related stem cells, while PDLSCs and DFPCs as periodontal-related stem cells [12,21,27]. In 2013, Marrelli et al. [34] described mesenchymal stem cells (MSCs) isolated from a human periapical cyst, which were termed hPCy-MSCs. Nevertheless, MSCs existing in the mesenchymal stem cell population in dental tissues are similar to one another and possess specific features relevant to each population they are isolated from [13,27,33,35]. Thanks to DSCs showing similarities to BMMSCs, their biological and functional properties have been intensely investigated to establish their potential value for regenerative medicine and tissue engineering [21,25,27,32,33,36,37]. The similarities and differences between DSCs obtained from different dental tissues in relation to BMMSC features are presented in Table 1.

One advantage of dental pulp tissue over the other human dental tissues is that it is significantly richer in stem cells, which may be isolated from the third molar and premolars extracted during orthodontic treatment or from deciduous teeth of young individuals. Consequently, hDPSCs and SHEDs became a promising field for future investigation and application in regenerative medicine and tissue engineering [12,13,21,27,28,35]. As presented in Table 1, dental MSCs have some features resembling MSCs, including fibroblast morphology, selective adherence to solid surfaces, formation of colonies in vitro, and expression of profile-specific markers similar to BMMSCs [13,25,27,35,38,39,40,41,42].

hDPSCs have some features resembling MSCs characteristics, including fibroblast morphology with selective adherence to solid surfaces and formation of colonies in vitro [27]. hDPSCs are characterized by their negative expression of hematopoietic antigens (e.g., CD14, CD19, CD24, CD34, CD45, HLA-DR) and positive of mesenchymal stem cells markers (e.g., CD10, CD13, CD29, CD44, CD73, CD90, CD105, CD106, CD117, CD146, STRO-1) [25,27]. Moreover, some of the pluripotent stem cell markers, such as Oct4, Nanog, Sox2, SSEA and c-Myc have been expressed in hDPSCs [43]. Apart from stemness markers, hDPSCs also express bone markers such as dentin sialophosphoprotein (DSPP), dentin matrix protein-1 (DMP-1), osterix (Osx), osteocalcin (OCN), osteopontin (OPN) alkaline phosphates (ALP) and type I collagen [13]. hDPSCs can be differentiated by modulation with growth factors, transcriptional factors, extracellular matrix proteins and receptors into mesodermal and nonmesodermal tissue cells including: osteoblasts, odontoblasts, adipocytes, chondrocytes, cardiomyocytes, neuron cells, corneal epithelial cells, hepatocytes and melanocytes [25].

Human SHEDs showed high expression of mesenchymal stem cell markers CD13, CD29, CD44, CD73, CD90, CD105, CD106, CD146, CD166 and STRO-1 but negative expression of the hematopoietic stem cell markers CD14, CD18, CD19, CD24, CD34, CD45. SHED has a high proliferation rate, high number of population doublings, and forms sphere-like clusters [26,27]. In vitro induction experiments demonstrated the potential of the SHEDs for osteogenic, adipogenic and neurogenic differentiation [44]. SHED has a stronger potential for self-renewal than DPSC [45].

hDFPCs are a group of dental mesenchymal stem cells that can be isolated from the dental follicle. hDFPCs express a series of classic MSC markers, including CD10, CD13, CD29, CD44, CD73, CD90, CD105, STRO-1, Notch-1, Oct3/4. Sox-2, Nanog1 and Nestin, are negative for hematopoietic stem cell markers such as CD11b, CD34, CD45 and HLA-DR [46,47]. Originating from the neural crest origin, hDFPCs can differentiate into osteoblasts, adipocytes, chondrocytes, cementoblasts and periodontal ligament cells as well as neuronal cells [47,48].

hSCAPs are located in the root apex of immature permanent teeth, have high proliferation rates, and a differentiation capacity in vitro similar to hDPSCs [49]. hSCAPs express the MSC markers CD13, CD29, CD44, CD73, CD90, CD105, CD106, CD146, CD166, STRO-1 and SCAP specific marker CD24. hSCAPs are negative for CD14, CD34, CD45, CD150 and HLA-DR expression. These stem cells are also plastic adherent and clonogenic and express low levels of odontoblast markers such as dentin sialoprotein (DSP), matrix extracellular phosphoglycoprotein [48]. They also have multiple differentiation potentials and can be induced into osteo/odontoblasts, lipoblasts and neuroblasts in vitro [49].

hGMSCs isolated from human gingival tissue revealed a fibroblast-like spindle morphology, and a colony-forming unit-fibroblast (CFU-F) [50]. GMSCs are positive for MSCs associated markers CD13, CD29, CD44, CD73, CD90, CD105, CD106, CD146, CD166, SSEA and STRO-1 and negative for hematopoietic markers CD14, CD34, CD45 and HLA-DR. hGMSCs possess some unique properties, particularly in have high proliferation activity and lower population doubling time than hBMMSCs [51]. hGMSCs have a high capacity to undergo osteogenic, chondrogenic and adipogenic differentiation [52].

hPDLSCs are heterogeneous, clonogenic, highly proliferative, and multipotent cells [40,41]. hPDLSCs are positive for typical MSCs surface markers such as: CD10, CD13, CD26, CD29, CD44, CD59, CD73, CD90, CD105, CD106, CD140b, CD146, CD166 but they also share pericytes markers such as STRO-1, Neuron-glial antigen 2 (NG2) and CD140b/platelet-derived growth factor receptor as well as neural crest derived cell markers: Nestin β-tubulin III CD271/p75NTR, SLUG and SOX10 [40,41]. hDPDLSCs are negative for: CD11b, CD14, CD19, CD31, CD34, CD40, CD45, CD79α, CD80, CD86 and HLA-DR [40,41]. hPDLSCs can differentiate into cementoblasts, osteoblasts, chondrocytes, and adipocytes. They can also differentiate into neurons, cardiomyocytes, endothelial cells, pancreatic islet cells, and corneal keratocytes [40,41]. hPDLSCs revealed immunomodulatory properties that can be performed by direct cell-to-cell contact or by the synthesis of specific bioactive factors that alter the phenotype of different immune cells [41]. For example, hPDLSCs could alter the proportion of T lymphocytes, increasing the proportion of T regulatory subsets over T helper-17 [41].

hPCy-MSCs showed a fibroblast-shape, adherence to plastic surface, high proliferative rate and self-renewal abilities [34]. The hPCy-MSCs population shares the same phenotypes with other types of oral derived MSCs and expresses stemness-related markers such as CD13, CD29, CD44, CD73, CD90, CD105, CD146 and STRO-1 [34,39]. These stem cells are negative for CD34, CD45 and HLA-DR expression [34,39]. hPCy-MSCs revealed high osteogenic/odontogenic and adipogenic differential potency [34,39].

Due to the evidence proving that—in contrast to the other dental tissues—the dental pulp is significantly richer with stem cells, hDPSCs and SHED became “a promised land” for future preclinical and clinical study [9,12,13,26,27,35]. Nevertheless, hDPSCs and SHEDs possess unique features different from BMMSCs [38]. Specifically, hDPSCs and SHEDs have a greater clonogenic potential and contain a higher number of stem/progenitor cells compared to BMMSCs [38]. Moreover, hDPSCs and SHEDs exhibit stronger neurogenetic and odontogenetic capabilities than BMMSCs; likewise, they have better neuroprotective properties and immunomodulatory abilities. For example, hDPSCs showed a higher suppression rate of T lymphocyte growth [12,38]. hDPSCs and SHEDs revealed a higher trophic factor expression compared to BMMSCs [12,35,53]. Finally, hDPSCs and SHEDs are considered MSCs, because they fulfill the standard ISCT requirements to define a cell type as MSCs [29,30]. BMMSCs, hDPSCs, and SHEDs, as multipotent adult stem cells isolated from different tissues, create fewer moral, ethical, or safety-related problems compared to embryonic stem cells, and are mostly investigated as a tool in regenerative medicine [3,8,35,36,53]. To date, BMMSCs, hDPSCs, and SHEDs have been demonstrated to be applicable in the treatment of various diseases, because they show immunomodulatory effects and a tissue repair ability [4,6,12,13,27,35,36,37,53]. As reported by Rodriguez-Fuentes et al. [16], by June 2020, there were over 1138 registered MSC clinical trials at www.clinicaltrials.gov (accessed on 1 November 2021) that used MSCs aspirated from bone marrow as therapeutic agents. BMMSCs were successfully used in the regeneration of damaged cartilage tissue [54,55], bone ligaments [56], and discs between the bones of the spine [57,58]. Only a few reports described clinical trials that used hDPSCs or SHEDs as a tool in regenerative medicine [9,42,59,60]. Autologous hDPSCs were successfully used in patients post-third molar extraction [59], with periodontal and bone defects [9,60], and in pulp regeneration [42]. Clinical studies suggest that hDPSCs and SHEDs are good candidates for use in regenerative medicine [9]. However, some authors pointed out that despite the encouraging results already obtained with in vivo experimental and clinical studies, MSC-based therapy conducted mainly with BMMSCs, hDPSCs, and SHEDs is still not considered to be the standard of clinical care [8,9,11,16]. Nevertheless, there is growing evidence that BMMSCs, hDPSCs, and SHEDs are effective in tissue regeneration and tissue engineering, but there are still considerable obstacles that limit their application in regenerative medicine [8,9,11,13]. It is worth underlining several challenges that have to be addressed to allow for a safe and efficient clinical use of MSCs in regenerative medicine [1,16,61]. The use of whole stem cells in therapy is limited by their proliferation potential, migration abilities, survival rate, differentiation complexity of cells in the host tissue, risk of viral transmission into the recipient, and teratoma formation [8,10,31]. MSC therapy is also limited by the absence of a standardized protocol for ex vivo expansion, culture conditions, and undetermined mode and optimal dosage of MSC infusion [62]. These limitations of MSCs used in regenerative medicine were suggested by Gnecchi et al. [63], who demonstrated that MSCs mediate their therapeutic effects through the secretion and release of trophic molecules known as secretomes. Data obtained by Gnecchi et al. [63] allow the theory to be revisited that the therapeutic applicability of MSCs in regenerative medicine is based on their capability of homing to the tissue injury site and differentiate into different functional cell types, leading to tissue repair. It is clear that additional factors are needed to optimize the therapeutic potency of MSCs in regenerative medicine. Further research by Teixeira et al. [20] provided more evidence that while transplanted MSCs do not commonly become part of the injured site in the repaired tissue, the secretome is the primary attribute of MSC-mediated repair and regeneration. The release of trophic and modulatory bioactive factors (secretome) by MSCs into the surrounding environment is what allows them to affect tissue homeostasis and promote tissue regeneration [11,17,19]. The above data along with growing experimental evidence show that the potential of MSCs to regenerate the injured tissue is fundamentally related to three mechanisms: “homing”, which refers to the ability of stem cells to migrate into the target organ/tissue due to chemical gradients [14]; their ability to self-renew and potential for multilineage differentiation; and a paracrine mechanism via the secretion of a broad array of bioactive factors [11,19,64].

## 3. Mesenchymal Stem Cell-Derived Secretome and Its Composition

MSCs can secrete or shed from their surface into the extracellular environment different growth and trophic factors, known as the secretome, which includes both the conditioned medium (CM) and extracellular vesicle (EV) fraction [10,11,17,19,64]. The MSC secretome has been defined as a spectrum of bioactive factors usually classified as soluble proteins and a vesicular fraction consisting of microvesicles (MVs) and exosomes (EXs), which are involved in the transference of various proteins and genetic materials [11,12,18,64].

As shown in Figure 1, the MSCs can release many different molecules as soluble factors and extracellular vesicles (EVs) (Meillana). Bioactive factors include various chemokines, cytokines, interleukins, growth factors, proteins, free nucleic acids (e.g., miRNAs, mRNAs, lncRNAs) and lipids (e.g., sphingolipids, cholesterol, ceramide) [35,65]. The secretome of cells is specific but changes in physiological or pathological conditions that are directly affected [11].

According to the size, composition and origin of EVs, they can be divided into three types: exosomes (EXs), microvesicles (MVs) and apoptotic bodies [11]. Soluble molecules usually exceed 10 nm in size [65]. EXs are small particles (40 to 120 nm in diameter), derived from multivesicular bodies [66,67]. EXs contain many bioactive macromolecules such as proteins, nucleic acids (e.g., miRNA, IncRNA, CircRNA, DNA) and lipids involved in biological regulation of cells [12,18,66,68].

EXs exert their action by delivering their contents directly into the cells without the need for specific receptor expression [11,12,18,67]. Depending on the cell sources EXs have a specific set of proteins and nucleic acids that can promote tissue regeneration by cells-cells communication [66,67]. EXs are involved in apoptosis regulation by immunomodulatory effects, anti-oxidative stress, and other actions [66]. EXs are rich in tetraspanins (CD9, CD63, CD81, CD86), annexins and heat shock proteins such as (HSP60, HSP70 and HSP90) [11,66]. Microvesicles (MVs) are heterogenous in size, ranging between 50 and 1000 nm in diameter and originating from the direct budding of the plasma membrane [11,65,67,69]. MVs contain nucleic acid, proteins and lipids [11,35]. MVs are pivotal for intercellular communication and can exert both paracrine and endocrine function [11]. The composition of EVs may change according to tissue and cell type of origin, as well as their physiological status [11,70]. Apoptotic bodies are very large vesicles (>1 µm) released after the fragmentation of the apoptotic cells [70,71]. Apoptotic bodies may contain many active bio-molecules and even intact organelles from apoptotic death cells [71].

## 4. Secretome of Dental Pulp MSCs and Their Biological Effect

It is worth mentioning that the MSC secretome has undergone intense research in MSCs isolated from oral tissue [12,18,21,53]. According to Sultan et al. [12], molecules secreted by hDPSCs or shed from their surface into the extracellular environment were previously regarded as waste containing cell debris. However, a later study revealed that the molecules secreted by hDPSCs and SHEDs into the conditioned medium (CM) showed a bioactive function and regenerative ability [12,13,35]. As with BMMSCs, different proteins can be released by hDPSCs and SHEDs in the CM or inside the EVs [12,35,53,72].

The secretome from MSCs is obtained in accordance with the same guidelines regardless of the source. In general, after stem cell culture in a serum-free medium with an antibiotic until 60–80% confluency (conditioning period), only the factors released by the cells as a supernatant (secretome/Conditioned Medium) are collected and stored at −80 °C after being snap-frozen to maintain the biological properties of the secretome [18,73,74]. To date, several methodologies have been tested to improve the MSC secretome production from different types of stem cells [8,11,17,18].

Figure 2 shows the methodologies use to improve the secretome production tested on BMMSCs which might be offered to hDPSCs and SHED-derived secretome production. Enhanced secretion of the therapeutic bioactive factors released by MSCs into the culture medium can be achieved through proteins, genetic and pharmacological manipulation of MSC culture, and activation with an apoptotic and inflammatory stimulus or even hypoxia and serum deprivation [8,17,18]. The impact of the basal culture medium on the production of the MSC secretome was observed by several authors [75,76,77]. It was found that the concentrations of cytokines and proteins released by MSCs depend on the level of glucose in the culture medium Dulbecco’s Modified Eagle Medium (DMEM) [75]. Some reports demonstrated that the use of a spheroid culture model improved the secretion of bioactive factors by MSCs compared to a monolayer culture [76,77]. The authors showed that spheroid cultures induced a higher protein secretion by MSCs, especially interleukin-1 (IL-1), vascular endothelial growth factor (VEGF), and granulocyte colony stimulating factor (GCSF), than a monolayer MSC culture [76,77]. It was found that exposing the MSCs to hypoxia altered their secretome properties, enhancing the production of bioactive factors [18,78]. Some studies found that growth factors, such as the VEGF-A, VEGF-C, placenta growth factor (PGF), granulocyte-macrophage colony-stimulating factor (GM-CSF), and inflammatory modulators (IL-1b, IL-6, IL1-15, and IL-1Ra), are upregulated in hypoxic conditions [78,79,80]. Promising results were obtained by pre-activating stem cells with signals to stimulate the sonic hedgehog (SHH) pathway. In this case, it was found that Wharton’s jelly-derived MSCs with SHH activation released a secretome that improved their angiogenic potential [81]. MSC secretome production can be improved with cellular extracts, including cues of the traumatic tissue, and by stimulating the cells with the extracts prior to in vivo application in the corresponding disease [82]. Modulation of adipose-derived MSCs with lipopolysaccharide increased the levels of the tumor necrosis factor alpha (TNF-α) and IL-6 and increased the regenerative profile of the secretome [83]. Releases by MSCs vesicles are easy to collect, and their therapeutic possibilities have also been analyzed [11,17,84,85]. A vesicle including EXs is usually isolated from stem cells according to the standard protocol. Stem cells are cultured for 48 h, and the conditioned medium is collected and centrifuged to eliminate dead cells and debris. Second centrifugation is carried out to eliminate the apoptotic bodies from the preparation. The supernatant is transferred to ultracentrifuge polyallomer tubes for ultracentrifugation at 10,000× *g* to pellet MVs and larger vesicles. The first ultracentrifugation at 100,000× *g* is used to pellet EXs and smaller vesicles, and the second ultracentrifugation is used to wash the isolated vesicles [86].

However, some differences between the biological activity of BMMSCs and dental MSC-CM were revealed. The dental pulp MSC-CM and SHED-CM showed higher neurotrophic, angiogenic, anti-apoptotic, and migration activity of the cells, neurite outgrowth, and immunomodulatory effects in vitro than MSCs isolated from bone marrow [35,87]. The hDPSC-CM and SHED-CM revealed an excellent nerve regenerative potential compared to the BMMSC-CM, which was associated with significantly higher colony formation and neurite extension, leading to appropriate neural differentiation and maturation [12,28,87]. The SHED-CM strongly promoted the expression of anti-inflammatory cytokine angiogenic and anti-apoptosis factors [88]. Similarly to the BMMSC-EVs, hDPSC- and SHED-EVs can be also subdivided into MVs and EXs [12,21,53]. EXs and MVs are pivotal for intracellular communication and can exert both paracrine and endocrine action [11,12,87]. MVs and EXs can also act as vehicles or stable transporters for bioactive molecules, such as cytokines and growth factors, transporting them from the producing cells to the adjacent or distant target cells by means of circulation [35,87].

The composition of the transported molecules depends on the producing cells [35]. Although the hDPSC and SHED secretomes are now being considered for regenerative medicine due to their paracrine effect mediated by the release of bioactive molecules, an important and frequently considered aspect of this type of therapy is its mechanism of action [12,35,38,53].

Figure 3 presents the spectrum of secretomes released by hDPSCs containing a broad range of bioactive soluble factors with anti-apoptotic and anti-inflammatory effects, angiogenetic regulation, and chemo-attractive and immunomodulation properties. These components include free nucleic acids, soluble proteins, EVs, and lipids [12,38]. The secretomes released by dental MSCs isolated from different oral tissues contain many common molecules [35]. As reported by El Moshy et al. [35], 1533 proteins were identified in the CM derived from bone marrow MSCs, adipose MSCs, and dental pulp MSCs using proteomic analysis; 999 proteins were contained in the CM of all three cell sources, of which 124 proteins were identified as the secreted extracellular proteins [35,89]. The secreted extracellular proteins found in the CM were suggested to be responsible for the regenerative effects of MSCs, such as angiogenesis, the inflammatory response, and cell migration [35,87,89].

In this respect, as shown in Figure 3, different fractions of the hDPSC and SHED derived secretome are undergoing thorough testing and analysis for their regenerative and protective properties. The dental pulp MSC-CM/EXs and SHED-CM/EXs contains immunomodulatory and anti-inflammatory factors, which play an important role in regulating the balance between the anti-inflammatory and pro-inflammatory properties of cytokines [12,35,38,53].

hDPSCs secrete strong immunomodulatory and anti-inflammatory cytokines, such as IL-6, IL-8, Transforming Growth Factor Beta (TGF-β), Hepatocyte Growth Factor (HGF), and Indoleamine 2,3-dioxygenase (IDO) [15,35]. TGF-β, HGF, and IDO are able to suppress the activation of T cells, the proliferation of peripheral blood mononuclear cells, or even allogeneic immune responses [15,38,53]. The secretion of IL-6, IL-8, and TGF-β by hDPSCs reduced the Toll-like receptor 4 (TLR-4) during neuroinflammation [28]. The dental MSC-CM induces an immunoregulatory activity by modifying the pro-inflammatory conditions and induces anti-inflammatory M2-like macrophage differentiation, thereby treating neural diseases [38,53,90]. A study on the immunomodulatory properties of hDPSCs observed that co-culture of hDPSCs and T cells promoted T cells to secrete vascular adhesion molecules-1, intracellular adhesion molecules-1, HGF, IL-6, IL-10, and TGF-β while reducing the secretion of pro-inflammatory cytokines, including IL-2, IL-12, IL-17A, and IL-6 receptor tumor necrosis factor α (TNF-α) [91]. Moreover, the dental MSC-CM suppressed the expression of pro-inflammatory cytokines, such as; TNF-α, IL-1β, IL-4, IL-6, IL-13, IL-17, IL-18, IFN-γ, COX-2, TLR-4, and NF-κB [12,13,15,35,92]. Studies reported that, when CD4+ T cells were cocultured with hDPSCs, the T cells demonstrated a high Treg expression achieved by blocking TGF-β1 and IL-10 signaling, resulting in a low Treg count, indicating that hDPSCs require stimulatory factors to exert their effects [38]. Likewise, the SHED-CM can induce anti-inflammatory effects by downregulating the expression of the proinflammatory cytokines IL-1β and tumor necrosis factor (TNF-α) and upregulating that of anti-inflammatory cytokines IL-4 and IL-10 [53]. Another mechanism with which hDPSCs spread their immunomodulatory and anti-inflammatory signals is their secreted EVs [12,35,53]. It was found that the administration of BMMSC or hDPSC-EVs can exert a protective effect by reducing inflammatory signals and has potential application in tissue regeneration [15,93]. Growing evidence suggests that hDPSCs and SHEDs may modify EV content in response to the environment, which shows that EVs are not only involved in extracellular signaling, but also offer innate biocompatibility and long-distance communication [18,94]. Another important aspect is the possible interaction between the different molecules present in the hDPSC-derived secretome that may act synergistically to potentiate the therapeutic effect in the regenerating tissue [15,53]. hDPSC-EVs reduced cytotoxicity through an anti-apoptotic mechanism by upregulating the Bcl-2 protein and downregulating the pro-apoptotic Bax protein in Aβ peptide-exposed human neuroblastoma (SH-SYSY) [95]. They also suppressed the inflammatory process by inhibiting TGF-β/Smad2/3 signaling and reducing the proliferation and cytotoxic potential of NK cells [96].

Several studies published recently revealed that the dental MSC-CM had a therapeutic effect through its anti-apoptotic function via the release of anti-apoptotic biomarkers [12,28,92]. Dental MSC-CM-based therapy significantly decreased the expression of pro-apoptotic proteins, such as Bax, p53, STAT, and cleaved caspase-3, but increased the expression of anti-apoptotic Bcl-2 in parenchymal cells, preventing their loss during inflammation [17,35,92,97]. It was shown that hDPSCs are able to inhibit TNF-α overexpression and maintain the level of the Bcl-xl protein, thus blocking the two apoptotic pathways, intrinsic and extrinsic [12]. To prevent apoptosis, hDPSCs secrete classic apoptosis inhibitor proteins belonging to the Bcl-xl [53].

Angiogenesis is strictly controlled by a delicate balance between stimulatory and inhibitory signals within the angiogenic process [12,28]. Dental MSC-CM promoted angiogenesis through the secretion of proangiogenic factors, including VEGF-A, FGF-2, platelet derived growth factor (PDGF), IGF-1, angiopoietin-2, MMP3, TGF-β, GM-CSF, G-CSF, IL-8, MCP-1, uPA, TIMP-1, and PAI-1, in addition to endogenous angiogenesis inhibitors (IGFBP-3 and endostatin) [12,35]. These factors play an important role in promoting new bone formation, dental tissue regeneration, and neuroregeneration [98]. The trophic factors expressed by stem cells are critical for vascular network remodeling; for instance, VEGF may be crucial in DSC-mediated vascular repair [15].

Accumulating evidence supports the neuroregenerative effect of the dental MSC-CM [12,32,99,100,101]. The crucial role of the dental MSC-CM as a modulator of the neurogenic microenvironment is to release multiple growth factors promoting neural growth and differentiation of neuronal cells [12,28,35]. Moreover, the dental MSC-CM contained factors involved in the reduction of neurotoxicity, such as VEGF, RANTES, FRACTALKINE, FLT-3, and MCP-1 and the Aβ-degrading enzyme neprilysin [28,35,38]. In animal experiments, DSCs provided cytoprotection through the secretion of neurotropic peptides, which contribute to neural repair and regeneration [88]. hDPSCs may play a neuroprotective function related to a high expression of trophic factors, including BDNF, GDNF, NGF, and NT-3, which promote the growth of neurons, induce neurogenesis at the site of injury, and play a pivotal role in neuroprotection against neurodegeneration and recovering neuron function [15,102]. Neurotrophins are a group of proteinaceous molecules that promote the development, function, and survival of neurons. These neurotrophic factors activate and bind to a family of receptor tyrosine kinases (TRKs). NT-3 binds to TrkC, BDNF to TrkB, and NGF to TrkA, sending a survival signal to neurons [10]. Another receptor, P75NTR, cooperates with the TRK family, transducing the signals from NGF, BDNF, and NT-3 to regulate a broad array of processes essential to the maintenance and development of the nervous system [12,15,38].

## 5. DSCs Secretome in Central Nerve System Diseases Therapy

Therapy using the dental MSC-derived secretome in neuronal diseases has been analyzed, but the results are still unsatisfactory, and the therapeutic potential in tissue regeneration, especially that of human dental MSC-derived secretome, is not yet confirmed [12,15,28,36].

**Alzheimer’s Disease (AD)** is an incurable neurodegenerative disease characterized by a decline in intellectual abilities and the appearance of β-amyloid plaques in the brain [95]. Considerable therapeutic effort has been made to treat AD or reduce its progression [15]. The dental pulp MSC-CM revealed a neuroprotective effect in an in vitro model of AD [28,35,53,95]. Interesting data were presented by Ahmed et al. [95] showing that the hDPSC secretome contains higher concentrations of VEGF, Fractalkine, RANTES, MCP-1, and GMCSF compared to those of bone marrow and adipose stem cells, which induce the anti-apoptotic effect. Moreover, neprilysin, as a membrane-bound protease detected in hDPSCs-CM, was strongly involved in degrading Aβ1 plaques [95]. Another protein secreted by hDPSCs is A2M, defined as a protease inhibitor and cytokine receptor that may play an important role in the neuroinflammatory response in AD patients [103,104]. The A2M cytokine is capable of binding to the β-amyloid and inducing its degradation [104]. The VEGF, RANTES, FRACTALKINE, FLT-3, and MCP-1 secreted by hDPSCs are also involved in the neuroprotective function and attenuation of neurotoxicity (Figure 4A) [95].

FRACTALKINE is considered as a key microglial pathway in protecting against AD-related cognitive deficits; RANTES increases neuronal cell survival and has a neuroprotective effect, while VEGF secretion enhances angiogenesis during tissue repair [10,95,105,106]. Moreover, growth factors and cytokines, such as VEGF, RANTES, FGF2, and Fractalkine can upregulate Bcl-2 protein expression, which enhances neuronal cell survival and reduces apoptosis caused by β-amyloid [95]. hDPSCs express high levels of BDNF and NGF, which are important positive neuroregulators in endogenous neuronal survival and synaptic plasticity (Figure 4A) [12,35]. Dominguez et al. [107] postulated that because CREB is a DNA-binding protein and acts as a transcription factor for BDNF, it is possible that a relationship exists between the role of BDNF expression and its regulation by CREB in restoring memory function in neurodegenerative disease. In a study on an animal model of AD, a high level of BDNF secreted by hDPSCs was associated with long-term memory, suggesting that CREB/BDNF is involved in the regenerative mechanism [35].

**Parkinson’s disease (PD)**, the second most common neurodegenerative disease worldwide, is caused by the loss of dopaminergic (DAergic) neurons in the *substantia nigra* resulting in a series of motor or non-motor disorders [108]. Drug therapy is the most effective and widely used treatment for PD patients, including administration of levodopa, DA agonists, amantadine, monoamine oxidase B (MAO-B) inhibitors, and catechol-O-methyltransferase (COMT) inhibitors [109]. Recently, hDPSCs and SHEDs have received extensive attention in PD therapy [86,108]. It is speculated that the neuroprotective effect of hDPSCs and SHEDs may be caused by the soluble factors they release, such as brain-derived neurotrophic factors, NGF and GDNF, which protect midbrain neurons damaged by 6-hydroxydopamine (6-OHDA), a selective dopaminergic toxin that induces apoptosis (Figure 4B) [86,110]. Zhang et al. [110] showed that the paracrine effect of neurotrophic factors secreted from SHEDs provides neuroprotection against neurodegeneration and recovery of the nigrostriatal dopamine neurons in model rats with 6-OHDA-induced degeneration.

Consequently, while the differentiation of stem cells into DAergic neurons is not the only purpose in PD cell therapy, the ability to produce and secrete neuroprotective factors may be more important [86,108]. Secreted neuroprotective factors may bind to specific receptors and trigger the activation of certain signalling pathways that coordinate cell function and survival [111]. In an in vitro model of PD, the SHED-CM demonstrated neuroprotective effects enhancing neurite outgrowth and repressed 6-hydroxydopamine-induced cell death [72]. Similarly, the SHED-CM showed a positive outcome in a PD rat model [108,112,113]. It was found that SHED secreted a large number of bioactive factors such as BDNF, GDNF, VEGF and IL-6, which mediated its immunomodulatory effect at the site [86,108]. Several reports confirmed the neuroprotective role of trophic factors presented in a SHED-conditioned medium in PD therapy [86,108,114,115]. Moreover, the secretion of cytokines IL-6 and TNF-α, combined with trophic factors, enhances immunoregulation and reverts the damage to host neurons. Collectively, the published reports indicated that the hDPSC- and SHED- conditional medium can be effective in neurogeneration of damaged neuron tissue [35,102,115,116].

**Spinal cord injury (SCI)** in humans can cause partial or complete loss of motor or sensory function [28,53]. Primary tissue disruption can be caused by mechanical damage to the nerve cells and blood vessels, and secondary injury can be caused by neuroinflammatory processes [28,102]. Because the hDPSCs-CM and SHED-CM secretomes provide a wide range of molecules with different properties, they demonstrated a remarkable neural regenerative potential with an ability to induce recruitment, neuronal maturation, and neurogenesis, and consequently, they were considered for SCI therapy [35,53,90]. It was observed that a high level of BDNF, GDNF, and NT-3 secreted by hDPSCs promoted motor functional recovery and inhibited cell apoptosis by reducing caspase-3 expression in SCI [28]. The hDPSC-CM has also been shown to reduce apoptosis and increase proliferation in resident Schwann cells following nerve injury [12,35]. The dental-MSC secretome may also affect the survival of the primary sensory neurons, which die because of nerve injury, by stimulating the endogenous survival Bcl-2 protein [94]. Early results provided by an experimental study performed by Martins et al. [117] confirmed that the binding of BDNF to TrkB induces the dimerization and subsequent autophosphorylation of the receptor. As with other receptor tyrosine kinases, TrkB can induce the activation of Erk and PI3K-Akt pathway BDNF/TrkB signaling, which is involved in neuronal differentiation, neuronal survival, axonal and dendritic branching, and axonal outgrowth and synaptic plasticity [118]. This Erk and PI3K-Akt signaling cascade promotes axonal outgrowth and axonal regeneration mediated by BDNF [10]. There are reports showing that the hDPSC-secretome is able to modulate phosphatidylinositol 3-kinase (PI3K)/Akt and MAPK pathways and the antioxidant effect of the superoxide dismutase 3 protein expression, thereby inducing neuron protection [35,102,119]. The network between different biomolecules secreted by dental MSCs-CM/EVs and signaling pathway receptors involved in the regenerative effect in SCI is shown in Figure 4C.

The dental MSC-CM contains molecules promoting vascularization, such as VEGF, FGF, HGF, EGF, which increases angiogenesis and the number of neurites, promotes the recovery of the spinal cord, increases the levels of ATP/NADH, and activates different survival pathways at the neuronal level, such as the phosphorylation of PI3K/AKT and JAK/STAT3, which promotes neuron growth (Figure 4C) [102,120]. VEGF released by hDPSCs or SHEDs may activate MAPK/ER1/2 signal transduction pathways, which promote neuron growth and affect the reorganization of the spinal motor network after SCI [18,121]. Furthermore, Kolar et al. [90] showed that the trophic factors secreted by hDPSCs promoted axon regeneration, despite the presence of axon growth inhibitors in a completely transected spinal cord model of SCI. The secretion of BDNF and NGF by hDPSCs reduced neurodegeneration in the early stage of neuronal apoptosis and enhanced motor and sensory neuron survival in SCI [102]. Moreover, the administration of the trophic factor secreted by hDPSCs in an animal model with SCI showed that these factors improved functional recovery and decreased the expression of TNF-α, IL-2, and IL-6 [122]. Dental MSC-derived secretomes are also able to reduce secondary inflammatory nerve injury and facilitate axonal regeneration, thereby reducing progressive hemorrhagic necrosis associated with the expression of the IL-1β ras homolog gene family member A (RhoA) and sulfonylurea receptor (SUR1) expression [123]. A neuroprotective role of hDPSCs-EVs in SCI was revealed [12,124]. A recent study demonstrated that a systemic injection of miR-133b-bearing MSC-EXs improved recovery from SCI by promoting the regeneration of axons through the activation of the survival Erk1/2 and STAT-3 signaling pathways in neurons (Figure 4C) [10,119]. In addition, MSC-EX may counteract the pro-inflammatory properties of A1 astrocytes by inhibiting nuclear translocation of the p65 subunit of NT-κβ, which is crucial for the generation of the inflammatory phenotype in these cells [53,125]. In a similar manner, via the downregulation of NF-κB p65 signaling, MSC-EVs reduced the migratory capacities of pericytes and maintained the structural integrity of the blood–spinal cord barrier [126]. hDPSCs exhibited a neuroprotective, anti-inflammatory, and angiogenic function when administrated unilaterally into the hind limb [124]. It was found that miR-133b-bearing-MSC-EX significantly improved the recovery of the hind limb locomotor function in an experimental animal model of SCI [119]. Studies showed that the molecular signaling pathway PI3K/AKT activated by lncRNA-F630028O10Rik enhanced microglial pyroptosis after activating the Toll-like receptor 4 (TLR4), which plays a key role in innate immune responses and activates the regeneration of the pro-inflammatory nuclear factor kB (NF-κB) in SCI [8]. Moreover, PI3K inhibitor EXs derived from MSCs can modulate the microenvironment of spinal cord lesions through their anti-inflammatory and pro-angiogenic effects induced by the suppression of inflammation and promote functional recovery following SCI [23]. Likewise, SHED-derived EXs and the SHED-CM improved the neurological outcome by inhibiting apoptosis in an in vitro dopaminergic neuronal model [35]. Furthermore, Huang et al. [127] showed that functional motor recovery following SCI was observed after a systemic application of MSC-EXs, which promoted neo-angiogenesis and suppressed TNF-α and IL-1β-driven inflammation by enhancing the production of immunosuppressive IL-10. In sum, these data clearly demonstrated that the hDPSC- and SHED-derived secretome can facilitate the regeneration of transected axons and induce functional recovery in SCI through multiple neuroregenerative mechanisms, such as angiogenesis and neurogenesis inhibiting the apoptosis of neural cells and replacing lost cells by differentiating stem cells into mature oligodendrocytes [15,128].

## 6. DSCs Secretome in Peripheral Nerve System Disorders Therapy

Peripheral nerve injury (PNI) may result in the loss of motor function, sensory function, or both, which leads to distal stump demyelination and degradation and may occur as a result of acute compression [12]. Experimental research confirmed the therapeutic effect of trophic factors released by hDPSCs and SHED in PNI and leading to neural regeneration [129]. The hDPSC secretome generates the neuroprotective microenvironment that prevents nerve degeneration and apoptosis and supports neurogenesis, axonal growth, re-myelination, and cell metabolism [12,35,112]. Research also indicates that dental pulp MSC-CM promoted the proliferation and migration of Schwann cells and inhibited their apoptosis, and promoted angiogenesis in an in vitro model of nerve injury [130]. Some authors suggested that the SHED–CM secretome contains BDNF, NGF, CNTF, GDNF, NT-3, and NT-4, inducing a more desirable extracellular microenvironment for peripheral nerve regeneration. The regenerative role of dental secreted MSC biomolecules is presented in Figure 4D [112,131]. It was shown that NGF released by the dental MSC-CM was involved in the differentiation and survival of sympathetic and sensory neurons and enhanced the migration of SCs in the peripheral nerve system, which effect was mediated through P75NTR [131]. Likewise, BDNF also strongly promoted axonal regeneration and functional recovery in a sciatic nerve defect, including enhanced axon growth, proliferation, migration, and survival [131]. Moreover, BDNF promoted the complexity of pyramidal neurons, with an increase in dendritic length in a layer-specific manner [130]. The SHED-CM containing MCP-1 and sSiglec-9 enhanced the neurite extension of the peripheral nerve [35,132]. This neuroprotective effect was evident through the promotion of migration, proliferation, and differentiation of Schwann cells, blood vessel formation, and nerve fiber extension [132]. On the other hand, MCP-1 and sSiglec-9 induced the polarization of the anti-inflammatory macrophage phenotype (M2), which started to express neuroprotective factors, such as BDNF, VEGF, IGF-1, NRG1, CNTF, and GDNF [15,28,35,113]. These trophic factors can modulate the proportions between the pro- and anti-inflammatory macrophage phenotype, thereby enhancing not only the survival of both resident neuronal and glial cells, but also participating in neurite chemoattraction to facilitate reinnervation [15,28]. Sultan et al. [111] revealed that soluble factors secreted from hDPSCs are able to stimulate the relevant neuron-associated gene expression in TGNC, increasing the expression of neural markers and axonal regeneration after cell isolation-induced injury. To increase neurovascularization, dental MSCs can also secrete VEGF, angiopoietin-1, IGF, PDGF, IL-6, IL-8, TGF-β, and HGF, which induces angiogenesis and promotes axonal growth and Schwann cell proliferation after PNI [112,133]. Growing evidence suggests that the hDPSC-CM and SHED-CM contain various cytokines, chemokines, trophic growth factors, with an ability to improve peripheral nerve regeneration and functional recovery [12,35,53,102].

## 7. Dental MSCs-CM in Tissue Regeneration

### 7.1. Repair of Bone Defects

Severe bone injury is difficult to heal by natural body processes, which is why clinical procedures must be used [134]. The repair rate of a bone defect depends on the size of the wound. Standard clinical approaches are restricted to autograft or allograft transplantation [134]. However, the successfulness of bone repair depends on the functional restoration of the adjacent tissue [134]. As an alternative solution in regenerative medicine or tissue engineering, the application of MSCs offers a wide range of therapeutic possibilities [134,135]. A majority of reports utilized an exogenous, autologous, or allogeneic MSC delivery into the injury site [9,35,42,56,59,60,116,136]. However, other studies attempted to recruit endogenous MSCs through the administration of the bioactive molecules released by BMMSCs, which may activate or inhibit different signaling pathways, enhancing the regeneration of bone defects [136,137]. As with BMMSCs, hDPSCs release factors that show a great regenerative potential [6,35,116]. Bone repair is directly associated with osteogenesis, which involves bone formation through osteoblasts [116,138]. This process is regulated by various pathways as presented in Figure 5, but not completely elucidated.

The BMP-4/Smad signaling pathway is essential for the osteogenic differentiation of hDPSCs, which can be suppressed by the tumor necrosis factor-inducible protein-6 (TSG-6) [139]. The osteogenic potency of hDPSCs is also regulated by WNT, which may subsequently affect JNK signaling pathways [140]. Moreover, the dental MSC-CM promotes osteogenesis by enhancing the migration and mineralization potential of stem cells through TGF-β1 and upregulating the expression of osteogenic genes, including runt-related transcription factor 2 (Runx2), osteocalcin (OCN), osteopontin (OPN), and osterix (Osx) [64,141]. Some studies found that mineralization occurs during bone regeneration linked with overexpression of TGF-β1/β2, BMP2, BMP4, MMP8, TUFT1, TFIP11, VEGF, and VEGFR2, which in turn is associated with the TGF-β-BMP signaling pathway activated by the dental MSC-secretome [142]. Moreover, the dental MSC-CM contains BMP7 and DSPP, which play a key role in bone formation and mineralization, as well as proteins regulating endochondral ossification (MINPP1), bone turnover (WISP2), and mineralization (enamelin) [143]. The VEGF released by hDPSCs at the site of tissue repair enhances bone tissue and blood vessel formation during distraction osteogenesis [35]. An alternative paracrine approach suggests that EXs secreted from dental MSCs improve osteogenesis and angiogenesis during bone repair in nonunion fractures, which has been attributed in part to the BMP-2 and VEGF signalling pathways [35,36,144]. Dental MSC-derived EVs enhance osteoblastic differentiation, improve osteochondral regeneration, and allow for bone defect-healing [21]. Haraki et al. [138] found that the SHED-derived CM had a higher therapeutic potential than stem cells in bone regeneration. Bone regeneration was more prominent with the CM, and mature bone formation and angiogenesis were observed only in the CM group. Interestingly, the SHED-CM was shown to contain factors correlated with angiogenesis, such as VEGF, and osteogenesis, such as OPG, OPN, BMP-2, and BMP-4 [138]. It is postulated that new bone formation is strictly related with angiogenesis [116]. Dental stem cells not only stimulate blood vessel formation via paracrine angiogenic factors, but also participate directly in angiogenesis by differentiating into endothelial cells (ECs) [116]. As shown in Figure 5, bone vascularization induced by the paracrine effects of hDPSCs is controlled by different molecular signaling pathways, such as p38/MAPK, PI3K/AKT, and MEK/ERK, which may activate or inhibit osteogenesis [145]. In an early stage of angiogenesis, dental MSCs can secrete a wide range of pro-angiogenic factors, including VEGF, FGF2, and PDGF, which bind to their corresponding receptors on ECs, leading to their proliferation [145]. Furthermore, it was shown that hDPSCs can functionally resemble perivascular supporting cells and induce more blood vessels when directly co-cultured with vascular ECs [146]. It was suggested that communication between these two types of cells is crucial for the induction of local neoangiogenesis [116]. The cellular communication between these two types of cells may be mediated by EXs [26,36,64]. According, some reports secreted by dental MSCs containing microRNA species, including miR-15/16, miR-31, miR-145, miR-221/222, miR-320a, miR-126, and miR-424, involved in the progression of the regenerative process [10,147].

### 7.2. Cartilage Tissue Damage Regeneration

Articular cartilage is an important weight-bearing tissue of the synovial joints. However, injury of the articular cartilage can cause osteoarthritis (OA) and seriously affect the physical and mental health of patients [148]. In the injured cartilage tissue, activated chondrocytes increase the production of cartilage, degrading catabolic enzymes, proinflammatory cytokines, and oxidative stress inducers, which induces the degradation of the cartilage matrix and chondrocyte apoptosis or necrosis via activation of the NF-κB pathway, leading to irreversible articular damage and functional impairment [149,150]. Standard surgical clinical techniques cannot regenerate the articular cartilage [151]. Due to the lack of blood vessels, nerves, and lymphatic vessels and the restriction of the dense extracellular matrix (ECM) on cartilage cells, the post-injury self-healing ability of the articular cartilage is very limited [148,151,152]. A new approach in cartilage-injury tissue therapy is regenerative medicine [148,151]. While MSCs in cartilage repair have been researched extensively, most studies indicate poor cell graft survival, suggesting the recovery of articular damage [68,153,154,155]. However, recent studies have shown that the stem cell secretome, such as the CM or EVs, can effectively mediate regeneration in the injured tissues, including cartilage [1,8,152]. Stem cells mediated their paracrine effect via secretion of bioactive factors, which induce the therapeutic process [8,152]. As shown in Figure 6, different growth factors, including TGF-β, FGF, VEGF, bone morphogenetic proteins (BMPs), and PDGF, were reported to have a beneficial effect on hyaline cartilage repair [149,150,152,156,157,158]. Ogasawara et al. [152] showed that SHED-CM treatment in an animal model promoted the regeneration and repair of temporomandibular joint osteoarthritis. The authors suggest that the SHED-CM not only inhibited the articular degradation cascade, but also regenerated the injured articular tissue by promoting the proliferation of the multipotent polymorphic cell layer and production of the cartilage matrix [152]. Furthermore, Muhammed et al. [156] found a protective effect of the SHED-CM on chondrocytes during osteoarthritis secretome therapy, with the stimulated chondrocytes displaying an enhanced anti-inflammatory ability, higher aggrecan and collagen type 12 expression, and significant MMP-13 and NF-κB downregulation. The authors pointed out that the SHED-CM significantly reduced the production of inflammatory cytokines, such as TNFα, IL-1α, and IL-6, and nitric oxide (NO), which are involved in cartilage degradation, and enhanced the production of IL-10, which protected the cartilage tissue from inflammation-related injury [152,156]. A similar observation was reported by other researchers, showing that the CM significantly decreased the levels of MMP-13 and IL-6 while increasing the level of TGF-β1 [22]. The SHED-CM secretes TGF-β, which promotes the differentiation of chondrocytes and the synthesis of collagen and proteoglycans, thereby maintaining cartilage homeostasis [150,159]. Moreover, MSC-CM can also inhibit the progression of OA by balancing the ratio of MMP-13 and TIMP-1 in cartilage, inhibiting chondrocyte apoptosis, and enhancing autophagy [160]. Ishikawa et al. [161] showed that an intravenously injected hDPSC-CM into the joint cavity in rheumatoid arthritis mice relieved joint symptoms and synovial inflammation. The histological evaluation of bone erosion and cartilage damage in the CM-treated animal group was significantly better than in the control group, and the gene expression of proinflammatory cytokines was significantly reduced [161]. The therapeutic potential of the hDPSC-CM was observed in another study showing that the hDPSC-CM may increase the proliferation and survival of immature murine articular chondrocytes in an in vitro model [150].

There are studies showing that MSCs-EVs play an important role in the trophic effect in cartilage regeneration [153,157,162]. Stem cell-derived EVs were reported to promote cartilage regeneration and prevent cartilage degeneration induced by OA [158,162]. Wu et al. [162] reported that adipose-MSC-derived EXs protected the articular cartilage from damage and improved gait abnormalities in OA mice by maintaining cartilage homeostasis, which may have been related to the inhibition of the mTOR autophagy pathway regulated by miR100-5p. Moreover, Zhao et al. [163] explored the effect of the MSC secretome on chondrogenic regeneration and showed that an application of miR-143 in an animal model caused the downregulation of the pro-inflammatory factors IL-6, TF-κB, and TNF-α and the overexpression of IL-10. However, the regeneration mediated by the molecular mechanisms in the injured tissue is not fully described [10,35,64]. Nevertheless, molecular studies on the EV-mediated mechanism show that miR-92a regulates the PI3K/AKT/mTOR signaling pathway by targeting noggin3, thus upregulating chondrocyte proliferation and matrix synthesis as presented in Figure 6 [164].

MSC-EVs may compensate for the reduced mitochondrial ATP production in osteoarthritis chondrocytes and increase their proliferation by activating survival Erk1/2 and AKT [163]. Erk1/2 and the AKT-derived pro-survival signal induced by MSCs-EVs in chondrocytes promote their proliferation, resulting in an enhanced regeneration of the OA cartilage [157].

## 8. Dental-MSCs and Their Secretome Combinated with Scaffold in Regenerative Medicine

Although stem cells are widely used in regenerative medicine, most of the preclinical and clinical studies demonstrated a poor functional integration into the host tissue, so several 3D scaffold-based cultures have been developed [9,12,28,58,59,60,61,62,165]. Stem cell 3D cultures are not without limitations; however, they enhance cell growth, differentiation, migration and improve cellular communications [21,166]. According to some authors, complete damage tissue regeneration might be achieved by using an appropriate scaffold, or associated with favorable biomolecules, such as growth factors, cytokines and combined with dental-MSCs [9,21,166]. Two main approaches for this combination are cell-based and cell-free tissue engineering. In the cell-based approach, stem cells are seeded and cultured onto the appropriate scaffold in vitro to produce the desired tissue before transplantation [9,39,166]. In the cell-free approach, a scaffold combined with growth and differentiation factors is embedded in the respective tissues, induces the homing of resident stem cells, and promotes their proliferation and differentiation [9]. In Ballini et al. [165] opinion interactions between stem cells and biomaterials play a crucial role in cell adhesion to scaffold surface and their response to differentiating stimulations. However, environmental cues, such as various growth factors/morphogens, markedly affect the behavior of DSCs seeded in scaffolds and are vital to the success of regenerative therapies [21,166]. However, there are data which indicated that material used in therapy might induce the inflammatory process, causing destruction of the surrounding tissue [167,168]. Recently, interesting data partly related to the process which might occur after engrafted materials in regenerative tissue were reported by Bressan et al. [168], who identify the role of titanium nanoparticles released from implants on the chronic inflammation and bone lysis in the surrounding tissue. The authors assessed the effect of titanium nanoparticles on MSCs physiology, reactive oxygen species (ROS) production and commitment onto osteogenic and adipogenic phenotype in patients affected by peri-implantitis [168]. The study showed the negative effect of titanium on surrounding tissue. High ROS production observed in all tissue specimens was strongly related to neutrophil recruitment and production of proinflammatory cytokines such as IL-1B, IL-6 and TNF. Moreover, RHOA/ROCK and WNT5A signaling pathways determining MSC commitment to the adipocyte or osteocyte lineages were up-regulated and promoted adipocytes differentiation and suppressed osteoblast differentiation [168].

In light of the above data, a scaffold used in regenerative medicine must be selected according to the biological and physical parameters. Good scaffolds need to show geometrical and mechanical properties that allow for oxygen and nutrient transport, enhanced cell survival, adhesion and promote cell differentiation and homing [21,39,165]. Tatullo et al. [39] analyzed the biological activity of mineral-doped CaSi-DCPD porous scaffold composed of dicalcium phosphate dehydrate (DCPD), and hydraulic calcium silicate (CaSi) combined with human periapical cyst-mesenchymal stem cells (hPCy-MSCs). Results showed a good growth rate of hPCy-MSCs and expression osteogenic/odontogenic markers such as dentin matrix protein-1 (DMP-1), and osteocalcin (OSC). The authors suggest that bioactive highly porous scaffold colonized by hPCy-MSCs may represent a promising strategy for regenerative healing of periapical and alveolar bone damage [39].

As presented in Table 2, several in vivo experimental studies described the biological effect of different scaffolds with loaded human dental-MSCs or their secretome in tissues regeneration including bone, cartilage tissues and spinal cord injury [138,169,170,171,172,173,174,175,176,177,178,179,180,181,182,183,184].

The used of hydroxyapatite (HA) scaffold combined with hSHED in repair bone defects was proven to be an effective agent in alveolar bone defect regeneration. The construct increased osteoprotegerin (OPG) and decreased tumor necrosis factor receptor superfamily, member 11a, NFKB activator (RANKL) expression, leading to a reduction of osteoclastogenesis [172]. Similar results found that hydroxyapatite (HA) matrix with poly(lactic-co-glycolic acid); PLGA enriched with seeded hDPSCs was used in therapy for bone damage. The PLGA with hDPSCs induced new bone formation and angiogenesis, leading to lesion size reduction in rabbits with bilateral, mandibular, critical-sized defects [169].

In another study, hDPSCs combined with three different scaffolds, made of L-lactide and DL-lactide (PLDL), a copolymer of DL-lactide (PDL) or HA/tri-calcium phosphates (TCP) were transplanted in mice model to establish their regenerative potential. The PDL scaffold seeded by hDPSCs showed the greatest dentin sialo-phosphoprotein (DSPP) expression, while the HA/TCP scaffold with hDPSCs showed the greatest dentin matrix protein-1 (DMP1) expression at 12 weeks after transplantation [173]. The authors indicated that PLDL, PDL and HA/TCP loaded with hDPSC seem to be a promising bioconstruct for odontogenic regeneration [173]. An experimental study presented by other investigators showed that alginate hydrogels containing BMP-2 with smaller pores and greater elasticity may induce greater bone regeneration [170]. It was found that human SHED encapsulated in hydrogel with a greater elasticity reduced expression of NF-κB p65 and Cox-2 in vivo model. The authors suggest that the mechanical features and microarchitecture of the scaffold influenced the fate of the encapsulated SHED and promote a proper environment in regenerated tissue [170].

Fahimipour et al. [174] developed a construct with heparin-conjugated collagen hydrogel immobilizing BMP-2, reinforced by 3D printed β-TCP based bioceramic scaffold. In rats implanted with the scaffold enriched with human SHED and heparin (HeP) or the scaffold with SHED but without heparin the status of mineralized tissue was similar but greater new bone formation was found when heparin was present. The study showed that BMP-2 presence increased the expression of genes involved in osteogenesis and induce appropriate cooperation between scaffold with loaded SHED and regenerated tissue [174].

In a study, where human SHED- and SHED-derived CM combined with atelocollagen sponge were implanted in animal model with calvarial bone defect, bone regeneration was evident in the bone defects treated with both SHED and SHED-derived CM [138]. Bone regeneration was dominated after SHED-derived CM treated resulted in better mature bone formation and angiogenesis. The authors suggested that biomolecules such as VEGF, OPG, OPN, BMP-2 and BMP-4 secreted by SHED have a paracrine effect on cells in adjacent tissue in a regenerated place and promoting angiogenesis and enhancing osteogenesis [138].

The regenerative effect of scaffold loaded by human dental stem cells and their secretome in periodontal defect was reported by Qiu et al. [175] who found that CM derived from hGMSCs and hPDLSCs loaded onto a collagen membrane were able to regenerate periodontal tissue and the results showed newly formed bone in the periodontal defects [175]. Moreover, hPDLSC-CM and hGMSC-CM reduced the inflammatory process by decreasing the levels of TNF- and IL-1 and increasing levels of IL-10, especially in the hGMSC-CM group facilitating tissue regeneration [175].

The osteogenic regenerative potential of a poly-(lactide) (3D-PLA) scaffold supplemented with human gingival MSCs and human gingival MSC-CM was analyzed in rat calvarial bone defects [176]. It was found that the growth factors and cytokines contained in the CM could activate mobilization and osteogenic differentiation of both endogenous MSCs and gingival MSCs by increasing genes expression involved in ossification and induced immunoregulatory effect on immunological components in surrounding tissue [176].

A study performed on mouse model revealed the regenerative potential of Tri-block PLGA–PEG–PLGA micro-spheres incorporated into a nanofibrous poly(l-lactic acid) PLLA scaffold loaded with EXs derived from hDPSCs in bone defect repair [177]. It was found that this construct stimulated bone tissue regeneration in vivo without the need for exogenous stem cell transplantation [177]. The authors observed that scaffolds containing EXs from mineralizing hDPSCs showed the best results, where a collagen-rich matrix, new bone tissue, and integration with the host tissue were observed [177]. Furthermore, the PLA scaffold enriched with hGMSCs-EVs, or polyethyleneimine (PEI)-engineered EVs (PEI-EVs), was used in calvarial bone tissue regeneration [178]. After implantation in rats subjected to cortical calvarial bone tissue damage, it was shown that the scaffold containing PEI-EVs, with or without cells improved bone healing and enhanced osteogenic properties [178]. Computed tomography (CT) revealed the formation of new bone spicules and blood vessels in rats’ calvarial bone defects implanted with 3D-PLA+PEI-EVs+human gingival MSCs and 3D-PLA+PEI-EVs. The authors hypothesized that the osteogenic potential of PEI-EV-human gingival MSCs loaded on 3D-PLA was mediated mainly by TGF-βR1, SMAD1, BMP2, MAPK1, MAPK14, and RUNX2 through TGF-β signaling pathway activation [178].

Similar results were observed by Pizzicannella et al. [179] who estimated the bone regeneration capacity of a collagen membrane enriched with human hPDLSCs and CM, EVs, or PEI-EVs in rats subjected to calvarial defects. Rats implanted with collagen membrane, hPDLSCs, and PEI-EVs showed increased bone regeneration in association with vascularization. The other results reported by Pizzicannella et al. [180] revealed a good regenerative effect of a construct formed by a PLA scaffold enriched with hGMSCs and their EVs in a rat calvarial defect model resulting in activation of bone regeneration and vascularization. The authors suggested that a positive effect might be associated with the upregulation of miR-2861 and 210 which might have a different paracrine effect in regenerated tissue, enhancing the regenerative properties of dental stem cells and reducing the inflammatory process [180].

The regenerative potential of a scaffold loaded with human dental stem cells was analyzed in spinal cord injury therapy [181,182,183]. In vivo study showed that grafts of the hDPSC/chitosan scaffold improved locomotor disability in animal models of SCI, by the secretion of BDNF, GDNF and NT-3, reducing the accumulation of active-caspase 3, and impairing axonal loss and degradation compared to the non-grafted animals [181]. Transplantation of hDPSCs combined with chitosan scaffolds into an SCI rat model resulted in the marked recovery of hind limb locomotor functions. The authors found that tissue loss, the number of apoptotic cells and axon degradation in the hDPSC/chitosan scaffold-transplanted group are significantly lower than those in the other experimental groups, whereas no lesions have been observed in the control group [181].

The used of Heparin-Poloxamer (HP), a thermosensitive material combining with hDPSCs in SCI model therapy, revealed that this construct improved the functional locomotor recovery in the SCI model by reducing apoptosis [182]. HP hydrogel had the ability to provide protective agents for loading and delivering biological macromolecules such as FGF and NGF, promoting axon regeneration and new blood vessel formation in injured site after SCI [182]. The functional recovery results showed that animals which were treated with HP-bFGF, HP-DPSCs, and HP-bFGF-DPSCs had a better outcome than the HP in SCI groups. Application of DPSCs combined with bFGF had a stronger impact on restoration and regeneration of neuronal function than bFGF-alone and hDPSC-alone applications [182]. In a related study, thermosensitive heparin hydrogels seeded by hDPSCs and containing bFGF efficiently reduce pro-inflammatory cytokine release in murine SCI and promote recovery of SCI [183]. The authors found that bFGF and hDPSCs worked together to attenuate tissue inflammation of the injured spinal cord, resulting in superior nerve repair [183]. This study proved that the in situ injection of bFGF and hDPSCs via a HeP hydrogel encouraged the neural regeneration responses in spinal cord injury. Reduced protein expression of MAP-2 and Ace-tubulin is attributed to neural death and dysfunction after SCI. Moreover, it was found that HeP-bFGF DPSCs played an important role in maintaining the microtubule structure, which indicated significant neurite repairing [183]. MRI scanning displayed various levels of neural repair by therapeutic interventions to the spinal cord tissue after SCI. The authors postulated that HeP-bFGF-DPSCs might reduce inflammatory process and showed the best restoration of the spinal cord tissue [183].

The usefulness of scaffolds combined with human dental stem cells for cartilage tissue injury therapy has been investigated [171,184,185]. Recently, it has also been reported that nanocellulose–chitosan thermosensitive hydrogel scaffolds carrying human DPSCs could promote cartilage formation both in vitro and in vivo [184]. The results showed the integrity of the thermosensitive hydrogel after injection in the desired site and prevention of infiltration of the cell load from the scaffold to neighboring tissues [184]. However, the 3D cell construct injected in the animal-induced host responded with an immune response demonstrated a positive impact on the biomaterial microenvironment. This immune response was in the form of an acute inflammatory reaction, which was resolved quickly by the end of the first week after injection. This immunological response attracted macrophages that, in parallel to hDPSCs, secreted growth factors and cytokines which are expected to support the regenerative activity of the bioconstruct [184]. Authors postulated that hDPSCs embedded in nanocellulose–chitosan thermosensitive hydrogel is a promising, minimally invasive, stem cell-based strategy for the regeneration of temporomandibular joint (TMJ) cartilage [184].

Mata et al. [171] investigated the effectiveness of hDPSCs combined with 3% alginate hydrogels in the regeneration of articular cartilage damage in a rabbit model. The scaffolds with hDPSCs were implanted in rabbits of a full-depth chondral joint defect. Alginate containing hDPSCs resulted in a clear regeneration of articular cartilage, which appeared to protrude into the articular cavity. This loss of cartilage was clearly diminished in animals implanted with alginate containing hDPSCs. Animals treated using alginate with hDPSCs exhibited marked regeneration of articular cartilage, characterized by the formation of new isogenic chondral groups and new chondral matrix compared with chondrocytes therapy [171]. The authors indicated that the administration of hDPSCs inhibited inflammatory processes in damaged tissue; they suggest that hDPSCs may be useful for the regeneration of articular cartilage injury [171]. hDPSCs seeded on chitosan-xanthan scaffolds as a tool in the therapy of cartilage damage tissue were investigated by Westin et al. [185]. The scaffold showed favorable characteristics for cartilage tissue engineering, such as high porosity, low cytotoxicity and mechanical properties, compatible with those characteristics of cartilage [185]. In vitro study showed that scaffold promoted hDPSCs adhesion, proliferation and differentiation into chondrocytes. In the authors’ opinion the chitoan–xanthian scaffold with seeded hDPSCs might be promising in the therapy of cartilage lesions [185].

The main question in regenerative medicine is if the scaffold combined with MSCs might generate an appropriate environment leading to restoration of damaged tissue. Most of the existing studies concerning the development of novel therapeutic approaches showed that combining stem cells with biomaterial scaffolds serves as a promising strategy for engineering tissue. However, advances in tissue engineering need to focus on combinations of scaffolds with DSCs and biomolecules which promote the regenerative capability of DSCs and enhance the therapeutic effect.

## 9. Research on Prospective Application for hDPSC- and SHED-Derived Secretome in Regenerative Medicine

Recently, a new approach in regenerative medicine and tissue engineering has involved the use of cell-free therapies in order to avoid any risks associated with the application of whole stem cells [9,10,11,17,35,108,186]. In the field of regenerative medicine, studies have shown that the use of the DSC-secretome has powerful regenerative potential and may be useful in the treatment of various diseases, including SCI and PNI, Alzheimer’s and Parkinson’s disease, as well as bone and cartilage tissue damage [12,15,35,86,95]. The subject literature contains numerous experimental reports on hDPSC- and SHED-derived CM/EVs used in therapy [187,188,189].

Table 3 presents several of the many reports on effective paracrine therapy in different diseases with the application of the human dental MSC-derived secretome. Human dental MSCs synthesize and release many biomolecules, which can be isolated and stored for long periods of time without loss of regenerative potency [11,12]. Although there are limited reports of the negative effects of dental MSC-CM/EVs in the regenerated tissue, it is worth underlining the benefits of the dental MSC-derived secretome in therapy. Firstly, there is no need to match the donor and recipient to avoid rejection [12]. Secondly, the use of CM for therapy reduces the risk of tumorigenicity and transmission of infection [11,12]. Thirdly, there are also financial and practical benefits due to a non-invasive cell collection procedure and lower cost and time of secretome production in accordance with GMP [11,12]. Furthermore, while the therapeutic application of the dental MSC-secretome has yielded promising results, there are still many obstacles to overcome before CM or EVs can be adopted in clinical procedures [190]. These include a lack of cGMP complaint protocols for the preparation of the MSC secretome, storage product, product stability, and quality control parameters that are essential to establish the safety and efficacy of MSC-CM. The main concern pointed out by Kumar et al. [143] remains in the inexistence of standard protocols for MSCs isolation/expansion and production. The basis of such uncertainly relies on the use of several medium conditions through the articulation of different culture parameters such as seeding cells density, oxygen tension, type of cells stimulation, pH, different types of culture systems (e.g., monolayer or spheroids) [65]. Considering that the dental MSC-secretome contains a wide range of different biomolecules, their production methods need to be standardized, and the activity of the biomolecules needs to be validated.

Another major concern in secretome therapy is the instability and short half-life of proteins and growth factors [11,186]. So far, there is still not a fulfilled and standard list of biomolecules or (mi)RNAs to be quantified which lead to selecting their own or appropriated molecules of interest, instead of considering secretome composition as a whole [76]. Moreover, gender, donor age and phenotype have been seen as important parameters contributing to the diversity of MSCs function and derivates. Additionally, during secretome production, it is important to achieve an appropriate balance between the stimulatory and inhibitory factors secreted by stem cells [186].

In sum, research conducted on the use of the dental MSC-derived secretome as a potential alternative to stem cell-based therapy makes it clear that the secretome has significant clinical potential. It is well documented that the therapeutic action is strongly mediated through paracrine factors released by human dental-MSC-derived secretome that include soluble factors and EVs, which apparently provide a supportive environment to host damaged tissue and enhance endogenous regenerative processes following injury or disease, promoting significant improvements when compared to cells therapy. The biomolecules released by hDPSCs and SHEDs may induce a paracrine effect that mediates cell-to-cell signaling for a long period of time during the regenerative process in damaged tissue. Nevertheless, the mechanism of dental MSC-CM or EV action is not fully documented; therapy based on the dental MSC-secretome seems to be a viable new option in regenerative medicine.

## Figures and Tables

**Figure 1 ijms-22-12018-f001:**
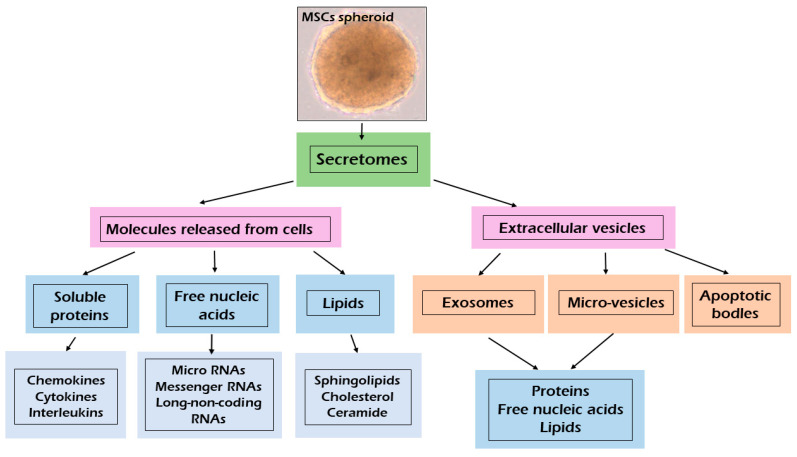
Composition of bioactive factors release by mesenchymal stem cells.

**Figure 2 ijms-22-12018-f002:**
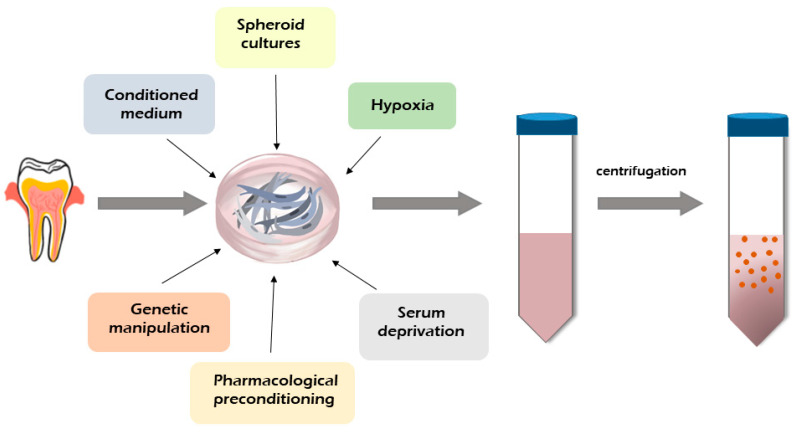
Dental MSC-derived secretome production and methods of its modification.

**Figure 3 ijms-22-12018-f003:**
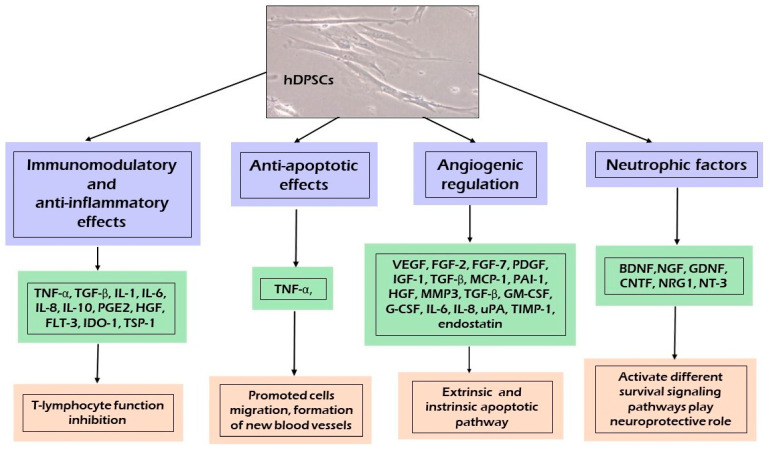
Paracrine effect induced by human dental pulp stem cells (hDPSCs) secretome.

**Figure 4 ijms-22-12018-f004:**
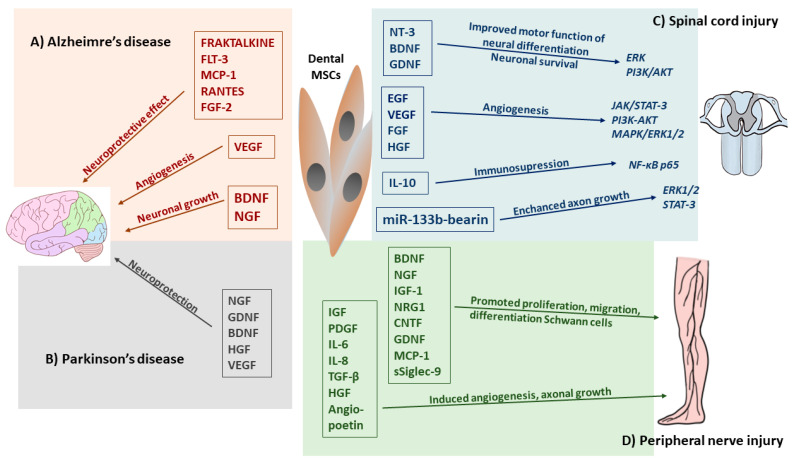
Effects of dental MSC-derived secretome in neurodegenerative diseases and central and peripheral nerve injury therapy. (**A**) Dental MSC-CM such as RANTES, FGF2, and Fractalkine enhanced neuronal cell survival and reduces apoptosis, BDNF, NGF involved in neuronal growth, whereas, VEGF promoted angiogenesis in Alzheimer’s disease. (**B**) Trophic factors (BDNF, NGF, GDNF, HGC, VEGF) and cytokines (TNF-α, IL-6) released by dental MCS-CM enhanced the regulatory function and reverts the damage sustained by host neurons in Parkinson’s disease. (**C**) Released by dental MSCs neurotrophic factors promoted neuronal differentiation and survival by activation of Erk, PI3/AKT and growth factors induced angiogenesis through JAK/STAT3, PI3K/AKT MAPK/Erk1/2 signaling pathways. Administration of dental MSC-EVs, miR-133 bearing promoted recovery form SCI by enhancing regeneration of axons through the activation of survival Erk1/2 and STAT-3 signaling pathways in regenerating neurons. Dental MSC-EVs induced immunosuppression at the site of SCI by enhancing production of IL-10 which suppressed neurotoxic A1 astrocytes through the inhibition of NF-κB-p65 signaling pathways. (**D**) Dental-MCS-derived secretome modulates nerve regeneration of peripheral nerve injury (PNI) by secretion of many neurotrophic factors which promoted survival of neuronal and glial cells, whereas, MCP-1 and sSiglec-9 molecules enhanced neurite extension of peripheral nerve. Cytokines and growth factors secreted by dental MSCs enhanced angiogenesis in PNI.

**Figure 5 ijms-22-12018-f005:**
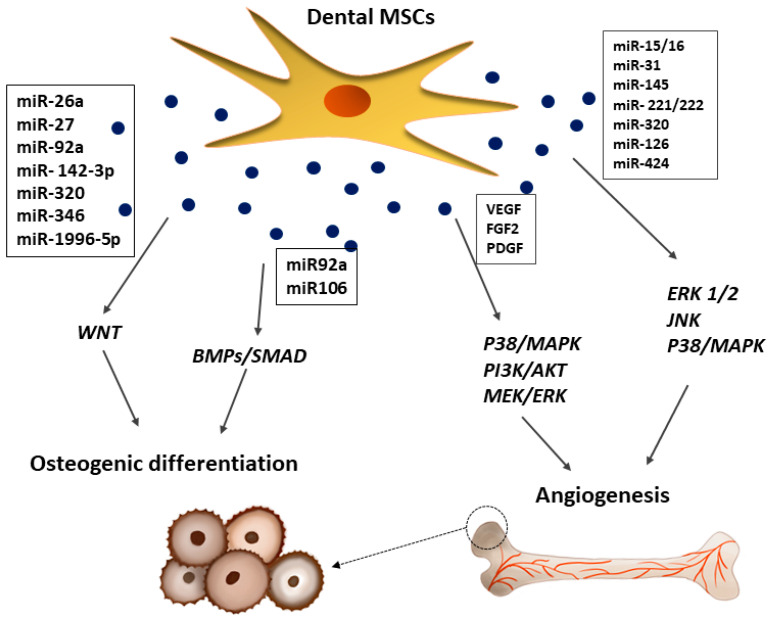
Effects of dental MSC-derived secretome in bone repair. Dental MSC-EVs promoted bone formation and osteoblast differentiation through the activation of BMPs/SMAD and WTN signaling pathways, whereas, VEGF and FGF2 released from dental MSCs promoted formation of new blood vessels by activation of p38/MAPK, PI3K/AKT, and MEK/ERK signaling pathways.

**Figure 6 ijms-22-12018-f006:**
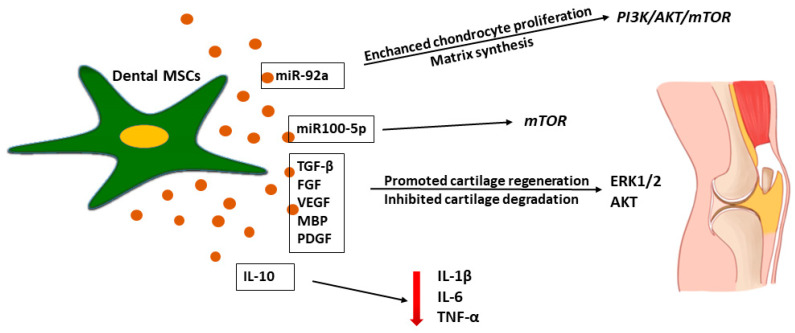
Effects of dental MSC-derived secretome in cartilage tissue regeneration. Dental MSC-MC inhibited inflammatory process in cartilage tissue injury by IL-10 secretion with reduced production of pro-inflammatory cytokines such as IL-1β, IL-6, TNF-α. Dental MSC-EVs promoted cartilage regeneration by administration of miR-92 which activated Erk1/2 and Akt-driven pro-survival signal in chondrocytes that promoted their proliferation resulting in enhanced regeneration of damage cartilage tissue.

**Table 1 ijms-22-12018-t001:** Biological features and multipotency capacity human mesenchymal stem cells from bone marrow and dental tissues.

	BMMSCs	hDPSCs	SHED	DFPCs	SCAPs	GMSCs	PDLSCs	hPCy-MSCs
Location	Bone Marrow	Permanent Tooth Pulp	Exfoliated Decidous Tooth Pulp	Dental Follcle Tissue	Apical Papilia	Gingival Tissue	Peridontal Ligament	Periaphical Cyst-Derived Mesenchymal Stem Cells
Morphological features:								
Shape in culture	Fibroblast-like cells	Fibroblast-like cells	Fibroblast-like cells	Fibroblast-like cells	Fibroblast-like cells	Fibroblast-like cells	Fibroblast-like cells	Fibroblast-like cells
Adherence to surfaces	+++	+++	+++	+++	+++	+++	+++	+++
Colony formation	++	+++	+++	+++	+++	+++	+++	+++
Proliferation potential	++	+++	+++	+++	+++	+++	+++	+++
Migration abilities	+	++	++	++	++	++	++	not done
Heterogeneity	++	++	++	++	++	++	++	not done
Clonogenic potency	++	+++	+++	+++	+++	++	+++	+++
Autologous application abilities	+++	+++	+++	+++	+++	+++	+++	+++
Immunoreactivity Positive biomarkers	CD10, CD13, CD44, CD73, CD90, CD105, CD106, CD146, CD166, STRO-1	CD10, CD13, CD29, CD44, CD73, CD90, CD105, CD106, CD117, CD146, STRO-1	CD13, CD29, CD44, CD73, CD90, CD105, CD106, CD146, CD166, STRO-1	CD10, CD13, CD29, CD44, CD59, CD73, CD90, CD105, STRO-1	CD13, CD24, CD29, CD44, CD73, CD90, CD105, CD106, CD146, CD166, STRO-1	CD13, CD29, CD44, CD73, CD90, CD105, CD106, CD146, CD166, STRO-1, SSEA	CD10, CD13, CD26, CD29, CD44, CD59, CD73, CD90, CD105, CD106, CD140b, CD146 CD166, STRO-1, STRO-3, STRO-4, NG2	CD13, CD29, CD44, CD73, CD90, CD105, CD146, STRO-1
Negative biomarkers	CD11b, CD14, CD19, CD34, CD40, CD45, CD79a, CD80, CD86, HLA-DR	CD14, CD19, CD24, CD34, CD45, HLA-DR	CD14, CD18, CD19, CD24, CD34, CD45	CD11b, CD34, CD45, HLA-DR	CD14, CD34, CD45, CD150, HLA-DR	CD14, CD34, CD45, HLA-DR	CD11b, CD14, CD19, CD31, CD34, CD40, CD45, CD79α, CD80, CD86, HLA-DR	CD34, CD45, HLA-DR
Multipotentialy:								
Osteogenic—	++	+++	+++	+++	+++	+++	+++	+++
Chondrogenic—	++	+++	+++	++	++	+++	++	not done
Neorogenic—	++	+++	+++	+++	+++	+++	+++	+++
Adipogenic—differentiation	++	+++	++	++	++	++	++	+++
Tissue repair, regeneration	bone, cartilage, neurodegenerative disease, spinal cord injury, myocardial regeneration	bone, cartilage, neurodegenerative disease, spinal cord injury, peripheral nerve injury, dental tissues, corneal regeneration	bone, cartilage, neurodegenerative disease, spinal cord injury, peripheral nerve injury, diabetes mellitus	bone, cartilage, peridontal tissue, neurodegenerative disease	bone, cartilage, peridontal tissue, neurodegenerative disease	bone, cartilage, peridontal tissue, neural disorders, skin injuries	bone, cartilage, peridontal tissue, neural disorders	bone, neurogenerative disease, oeridontal tissue

+++ high, ++ moderate, + low.

**Table 2 ijms-22-12018-t002:** The regenerative effect of scaffold-based combined with human dental stem cells and scaffold enrichment with dental MSC-derived secretome used in vivo studies.

Author (Publication Year)	Source of Human Stem Cells	Biomaterials	Animal Model	Results
Prahasanti et al., 2019 [172]	SHED	hydroxyapatite (HA) scaffold	Alveolar bone defect model Wistar rats	Improved alveolar bone defect regeneration
Gutierrez-Quintero et al., 2020 [169]	DPSCs	HA matrix with polylactic polyglycolic acid; (PLGA) scaffold	Bilateral mandibular critical-sized defects New Zealand rabbits	Induced new bone formation and angiogenesis. The scaffold without hDPSCs was less efficacious
Atalayin et al., 2016 [173]	DPSCs	L-lactide and DL-lactide; (PLDL), copolymer of DL-lactide; (PDL), and HA/tri-calcium phosphates; (TCP) scaffold	Subcutaneous implantation Immunocompromised mice	PLDL, PDL, and HA-TCP enriched with hDPSC seemed to be promising scaffolds for odontogenic regeneration
Ansari et al., 2017 [170]	SHED	Alginate hydrogels containing BMP-2, scaffold	Subcutaneous C57BL/6 mice	Scaffold with smaller pores and greater elasticity was found to potentially induce greater bone regeneration
Fahimipour et al., 2019 [174]	DPSCs Enrichment BMP-2	Heparinconjugated collagen (Col) hydrogel reinforced by 3D printed-TCP-based bioceramic scaffold	Subcutaneous implantation Male Fischer 344 rats	A greater new bone formation was found when heparin was present. BMP-2 increased the expression of genes involved in osteogenesis
Hiraki et al., 2020 [138]	SHED-CM	Atelocollagen sponge	Calvarial bone defect model. Deficient mice (BALB/c-nu)	Enhanced bone regeneration and angiogenesis
Qiu et al., 2020 [175]	GMSC and PDLSC-CM	Col membrane	Periodontal defect model. Wistar rats	Newly formed bone and reduced inflammation
Diomede et al., 2018 [176]	GMSCs -CM	PLA scaffold	Calvarial defect. Wistar rats	Induction of new bone formation and osseointegration
Swanson et al., 2020 [177]	DPSCs- EXs	Tri-block PLGA–PEG–PLGA micro-spheres incorporated into a nanofibrous PLLA scaffold	Calvarial defect. C57BL/6 mice	Bone tissue regenerated
Diomede et al., 2018 [178]	GMSCs-EVs, or PEI engineered EVs	PLA scaffold	Calvarial defect. Wistar rats	Improved bone healing by showing better osteogenic properties
Pizzicannella et al., 2019 [179]	PDLSCs- CM, EVs, or EVs engineered with PEI	Col membrane	Calvarial defect. Wistar rats	Increased bone regeneration in association with vascularization
Pizzicannella et al., 2019 [180]	GMSCs- EVs	PLA, scaffold	Calvarial defect. Wistar rats	Bone regeneration and vascularization were observed
Zhang et al., 2016 [181]	DPSCs	Chitosan scaffolds	SCI rat model	Transplantation of hDPSCs together with chitosan scaffolds into an SCI rat model resulted in the marked recovery of hind limb locomotor functions.
Luo et al., 2018 [182]	DPSCs- FGF	heparin-poloxamer (HP) hydrogel	SCI rats model	HP-bFGF-DPSCs had a significant impact on spinal cord repair and regeneration
Albashari et al., 2020 [183]	DPSCs-bFGF	heparin (HeP) hydrogel	SCI mouse model	vivo application of HeP-bFGF-DPSCs regulated inflammatory reactions and accelerated the nerve regeneration through microtubule stabilization and tissue vasculature. Prevented microglia/macrophage activation
Talaat et al., 2020 [184]	DPSCs	Nanocellulose–Chitosan Hydrogel (NC-CS/GP-21)	Subcutaneous injections. Sprague Dawley rats	hDPSCs/NC-CS-GP-21 scaffold induced the remodeling and regeneration of damaged cartilage
Mata el al., 2017 [171]	DPSCs	alginate hydrogels	cartilage damage rabbit model	significant cartilage regeneration, formation of new isogenic chondral groups and new chondral matrix

**Table 3 ijms-22-12018-t003:** The regenerative effect of human hDPSC and hSHED-derived secretome used in experimental studies.

Author (Publication Year)	Source of Stem Cells	Secretome Composition	Disease or Target Tissuse	Paracrine Effect
Imanishi et al., 2021 [36]	Human DPSC-EVs	CD9	Bone defect repair Animal model	hDPSC-EVs promoted new bone formation.
Ahmed et al., 2016 [95]	Human DPSC-CM	VEGF, FLT-3, RANTES, fractalkine, GM-CSF, MCP-1, neprilysin	Alzheimer’s disease In vitro model	hDPSC-CM inhibited apoptosis in neuroblastoma cell line and increased its viability
Narbute et al., 2019 [115]	Human SHED-EVs	80 proteins in EVs derived from SHED culture were identified	Parkinson’s disease Animal model	Suppression of gait impairments and normalization of tyrosine hydroxylase expression
Hiraki et al., 2020 [138]	Human SHED-CM	OPG, TIMP-1, ANG, MCP-1, IL-6, OPN, BDNF, NT-3, HGF, EGF, M-CSF, FGF-2, GDNF, β-NGF, PDGF-β, IGFBP-3, BMP-2, BMP-4, VEGF-A and VEGF-C.	Calvaria defect inflammatory bone loss Animal model	SHED-CM enhanced bone volume and formation
Muhammad et al., 2020 [156]	Human SHED-CM	TGF-β1, IL-6, IL-10	OA chondrocytes In vitro model	SHED- CM increased the expression of aggrecan and COL 2 in OA chondrocytes. Moreover, CM regenerate extracellular matrix proteins and mitigate increased MMP-13 expression through inhibition of NF-κB in OA chondrocytes due to the presence of bioactive molecules.
Ishikawa et al., 2016 [161]	Human SHED-CM	HGF, IL-12, furin, IL-1RA, RAGE, OPG, MCP-1, sSiglec-9	Rheumatoid arthritis In vitro model	SHED-CM promoted M2 anti-inflammatory state and inhibited osteoclastogenesis
Swanson et al., 2020 [177]	Human DPSC-EXs	CD9,CD63, CD81, TSG 101 proteins	Calvarial defect Animal model	hDPSC-EXs induced bone tissue regeneration
Mita et al., 2015 [187]	Human SHED-CM	Ym-1, Arginase-1, Fizz1, IL-10, mRNA of BDNF, NGF, IGF	Alzheimer’s disease Animal model	SHED-CM attenuated the proinflammatory response induced by β-amyloid plaques
Matsubara et al., 2015 [188]	HumanDPSC-CM	M2- like macrophage inducers: Siglec-9, IL-6, MCP-1	Spinal cord injured Animal model	hDPSC-CM suppressed inflammatory process, promoted functional recovery of neurons
Luo et al., 2019 [189]	Human SHED-EVs	CD9, CD63, TSG101, MiR-100	Osteoarthritis (OA) In vitro model	Suppression of inflammation in TMJ osteoarthritis
Shen et al., 2020 [191]	Human DPSC-EVs	DPSC-EXs expressed CD9, HSP70, TSG101	Periodontitis Animal model	hDPSC-EVs promoted regenetaion
Jin et al., 2020 [192]	Human DPSC-EVs	CD9, CD63, CD81, TSG 101	Bone defects repair Animal model	hDPSC-EVs promoted the regeneration of bone defects

## Data Availability

Not applicable.

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
