# Peer review of "Dental Pulp Stem Cell-Derived Secretome and Its Regenerative Potential"

_ijms, 2021, doi:10.3390/ijms222112018_

Round 1

Reviewer 1 Report

Authors have submitted “Dental pulp stem cell-derived secretome and its regenerative potential ”: they aim to outline the current data on the DPSC and SHED-derived secretome as a potential candidate in the regeneration of bone, cartilage, and nerve tissue.

Some major issues should be clarified before any further consideration:

- the authors should also discuss about the role of secretome and the crosstalk among cells with local environment like biomatrices and bioscaffolds (Ballini A, Boccaccio A, Saini R, Van Pham P, Tatullo M. Dental-Derived Stem Cells and Their Secretome and Interactions with Bioscaffolds/Biomaterials in Regenerative Medicine: From the In Vitro Research to Translational Applications. Stem Cells Int. 2017;2017:6975251).

-Moreover, the authors lacked to report about the clinical interaction between bioscaffolds and tissues in tissue engineering and organs repair, and the related role of secretome (Tatullo M, Spagnuolo G, Codispoti B, Zamparini F, Zhang A, Esposti MD, Aparicio C, Rengo C, Nuzzolese M, Manzoli L, Fava F, Prati C, Fabbri P, Gandolfi MG. PLA-Based Mineral-Doped Scaffolds Seeded with Human Periapical Cyst-Derived MSCs: A Promising Tool for Regenerative Healing in Dentistry. Materials (Basel). 2019 Feb 16;12(4):597.”).

- Finally, Secretome is not only a topic depending from cell releasing of topical factors, but also from other ectopic co-factors that could interact/alter tissue adaptation after a repeated micro-trauma (e.g. Please, see and discuss: “Bressan E, Ferroni L, Gardin C, Bellin G, Sbricoli L, Sivolella S, Brunello G, Schwartz-Arad D, Mijiritsky E, Penarrocha M, Penarrocha D, Taccioli C, Tatullo M, Piattelli A, Zavan B. Metal Nanoparticles Released from Dental Implant Surfaces: Potential Contribution to Chronic Inflammation and Peri-Implant Bone Loss. Materials (Basel). 2019 Jun 25;12(12):2036.”).  

- The authors have described dental-derived stem cells; however, they lack to include human-periapical cysts-mesenchymal stem cells: authors must improve their paper with this information, including tables and discussion.

- General description of mesenchymal stem cells is not useful and should be shortened; on the other hands the authors should better describe /with some more figures involved/ the figures 1 and 2

Minor issues

- Limitations should be better reported and discussed.

- Conclusions need to be increased with more related text and future insights on this matter.

Reviewer 2 Report

Title of Manuscript: Dental pulp stem cell-derived secretome and its regenerative potential

Comments to the Author(s):

The authors reviewed the secretome derived from MSCs, especially from DPSCs. This article is important because the function of stem cell-derived secretome is diverse and many researchers are now investigating their characteristics. Some comments are shown below.

  1. L13; The author should put a hyphen after DPSC.
  2. L16; The author should put a comma after cartilage defects.
  3. 1; The description of stem cell characteristics does not match the results. The authors should carefully check the table.
  4. PDLSCs express more markers than the markers that the authors showed. They should add more information for PDLSCs referring to the article, Periodontal Ligament Stem Cells: Regenerative Potency in Periodontium, Tomokiyo A, et al. Stem Cells Dev. 2019.
  5. L122 et al.; The authors use the words hDPSC and DPSC, which are very confusing. Does DPSC mean non-human DPSC? If so, they should show the type of animal before DPSC.
  6. L187; The authors should use EXs and MVs instead of exosomes and microvesicles because they define these abbreviations before.
  7. L190; The authors use the word CM as various meanings. They should define the meaning of CM correctly.
  8. L199; The word DMEM is used only once, therefore the authors do not need to use abbreviation for it. The article includes abbreviations that need to be corrected. Please check them carefully.
  9. L207; The word VEGF is defined before, therefore vascular endothelial growth factor-A should be removed.
  10. L207; The shape of the parentheses after PGF is strange.
  11. L283; The cluster of differentiations is usually arranged in numerical order.
  12. L294; Interleukins are usually arranged in numerical order.
  13. L294; Interleukin is defined before, therefore the authors should use abbreviation for it. Also, abbreviations are sometimes not used correctly after their definition. Please check them carefully.
  14. L362; The authors use p75NTR instead of P75NTR.

Round 2

Reviewer 1 Report

none to add